# Total synthesis of mycobacterial arabinogalactan containing 92 monosaccharide units

Yong Wu[1], De-Cai Xiong[1], Si-Cong Chen[1], Yong-Shi Wang[1] & Xin-Shan Ye[1]

Carbohydrates are diverse bio-macromolecules with highly complex structures that are involved in numerous biological processes. Well-defined carbohydrates obtained by chemical synthesis are essential to the understanding of their functions. However, synthesis of carbohydrates is greatly hampered by its insufficient efficiency. So far, assembly of long carbohydrate chains remains one of the most challenging tasks for synthetic chemists. Here we describe a highly efficient assembly of a 92-mer polysaccharide by the preactivation-based one-pot glycosylation protocol. Several linear and branched oligosaccharide/polysaccharide fragments ranging from 5-mer to 31-mer in length have been rapidly constructed in one-pot manner, which enables the first total synthesis of a biologically important mycobacterial arabinogalactan through a highly convergent [31 + 31 + 30] coupling reaction. Our results show that the preactivation-based one-pot glycosylation protocol may provide access to the construction of long and complicated carbohydrate chains.

[1] State Key Laboratory of Natural and Biomimetic Drugs, School of Pharmaceutical Sciences, Peking University, Xue Yuan Road No. 38, Beijing 100191, China. Correspondence and requests for materials should be addressed to X.-S.Y. (email: xinshan@bjmu.edu.cn).

Carbohydrates are involved in many key biological processes, such as cell signaling, cell proliferation and differentiation and viral and bacterial infections, as well as immunoresponse[1–3]. Naturally occurring carbohydrates and glycoconjugates usually exist in microheterogeneous forms, making the isolation of pure carbohydrates and glycoconjugates from natural sources difficult or even impossible in most cases. Therefore, chemical synthesis becomes the main approach to obtain well-defined carbohydrates[4–7]. However, unlike peptides and oligonucleotides, which can be routinely prepared by automated solid-phase synthesizers, the oligosaccharide synthesis is much more difficult. The major challenge for oligosaccharide preparation is the regio- and stereochemistry[8,9] issues in each glycosidic bond formation, making oligosaccharide synthesis a tedious and time-consuming process. Therefore, oligosaccharide synthesis becomes a daunting task, especially when polysaccharides are chosen as the target molecules. Indeed, only a few examples of the synthesis of oligosaccharide sequences containing >20 units have been reported over the past few decades[10–18]. These syntheses are challenging because multiple steps of protective group manipulation and intermediate purification are required in most cases.

Arabinogalactan is an essential structural constituent of mycobacterial cell wall, which plays critical roles in the infectivity and pathogenicity of *Mycobacterium tuberculosis*[19]. Based on experiments and analyses[20–22], the primary structure of arabinogalactan has been established as a linear galactan composed of about 30 alternating $\beta$-$(1\rightarrow5)$-linked and $\beta$-$(1\rightarrow6)$-linked D-galactofuranose (Gal$f$) residues, to which up to two[22] highly branched arabinan chains (each containing 31 D-arabinofuranose (Ara$f$) residues) are attached. The arabinogalactan motifs are useful probes for investigating the biosynthesis of mycobacterial cell wall, especially for characterization of the enzymes that process this polysaccharide, and those enzymes are attractive targets for the development of new antituberculosis drugs[23]. To this end, some solution phase[14,15,24–27] and automated solid-phase[28] strategies have been developed for the assembly of motifs up to 22 residues, among which the Lowary group[14] and Ito group[15] have elegantly synthesized the docosasaccharide arabinan motif via the convergent $[5+5+12]$ and $[7+7+8]$ coupling strategies, respectively. While almost all these syntheses relied upon the stepwise synthesis of oligosaccharide fragments, the object of this study is to achieve the total synthesis of the whole complex polysaccharide rather than the truncated fragments in an efficient way.

In the preactivation-based one-pot glycosylation strategy, several glycosyl donors are allowed to react sequentially in the same vessel regardless of the anomeric reactivities, generating a single oligosaccharide as the main product, which can significantly simplify the synthetic process and increase the overall efficiency[29]. Herein, by utilizing the preactivation-based one-pot glycosylation protocol, we report the first total synthesis of a biologically important mycobacterial arabinogalactan composed of 92 monosaccharide units. Our synthetic strategy involves: (1) several scalable one-pot coupling reactions to generate the linear and branched oligosaccharide fragments, (2) the stereoselective $\beta$-arabinofuranosylation by preactivation protocol, (3) the further one-pot coupling reactions of oligosaccharide fragments for the rapid assembly of polysaccharides up to 31-mer, and (4) the convergent $[31+31+30]$ coupling reaction for the final construction of the target polysaccharide.

## Results

**Retrosynthetic analysis**. The target polysaccharide arabinogalactan **1** was disconnected into two sizeable fragments, that is, the linear Gal$f_{30}$ acceptor **2** and the branched Ara$f_{31}$ donor **3** (Fig. 1). It was conceived that Gal$f_{30}$ acceptor **2** would be rapidly assembled via a five-component one-pot coupling of several oligosaccharide fragments **4–7**. As for the synthesis of Ara$f_{31}$ donor **3**, oligosaccharide fragments **10–12** were designed to carry out a four-component one-pot glycosylation reaction. For the preparation of heptasaccharide **10**, thioglycoside donors **13a–c** and thioglycoside acceptor **14** were planned for the construction of the challenging $\beta$-arabinofuranosyl linkages. Finally, it was expected that all the oligosaccharide fragments (**8**, **9**, **15–17**) would be accessible by the preactivation-based one-pot oligosaccharide synthesis starting from various monosaccharide building blocks. Overall, it was anticipated that the major challenge of our plan towards the total synthesis of arabinogalactan **1** would rely on the efficiency of one-pot glycosylation reactions, especially when large oligosaccharide fragments were attempted as the components in one-pot coupling reactions.

**Synthesis of Gal$f_{30}$ acceptor 2**. To test our one-pot strategy for oligosaccharide synthesis, three monosaccharide building blocks **18–20** were designed and synthesized (Supplementary Fig. 1). Using these building blocks, the assembly of hexasaccharide **8** in a six-component one-pot manner ($\mathbf{18+19+20+19+20+19}$) by preactivation protocol was tried, which should be rather challenging as up to five glycosidic linkages must be correctly constructed. To our delight, when promoted by stoichiometric $p$-toluenesulfenyl chloride/silver triflate ($p$-TolSCl/AgOTf)[29], all glycosylation steps underwent smoothly and none of the side products interfered with the reaction. After optimization of the reaction conditions, hexasaccharide **8** was obtained in 63% overall yield and on a perfect scale (1.07 g) within several hours (Fig. 2a). The desilylation of **8** provided **5** (85% yield), which was re-protected with benzoyl group to give **4** in 96% yield. Subsequently, the coupling reaction of **8** with 1-octanol afforded **21** (91% yield), which was followed by desilylation to provide **6** in 87% yield.

With hexasaccharides **4–6** in hand, the further iterative one-pot glycosylation was performed. The five-component one-pot coupling of these oligosaccharides ($\mathbf{4+5+5+5+6}$) was realized successfully, producing the 30-mer polysaccharide **22** in 68% overall yield (Fig. 2b). It was noteworthy that some deletion sequences were difficult to be removed by column chromatography on silica gel. Gratifyingly, given the difference in molecular weight between the deletion sequences and desired product, size exclusion chromatography was then used to obtain the pure 30-mer polysaccharide **22** (Supplementary Fig. 2). The identity of **22** was confirmed by its nuclear magnetic resonance (NMR) and matrix-assisted laser desorption/ionization–time of flight (MALDI-TOF) mass spectra (see Supplementary Information for details). Finally, the global deprotection of **22** via successive debenzoylation and debenzylation provided the 30-mer galactan **23** ($[M+Na]^+$ $m/z$ calcd. for 5017.4, found: 5018.1).

Having established a highly efficient one-pot approach to the synthesis of polysaccharide up to 30-mer, we turned to accomplish the construction of Gal$f_{30}$ acceptor **2**. As shown in Fig. 1, an additional hexasaccharide **9** equipped with two levulinoyl groups was needed. Initially, monosaccharide **24a** (Supplementary Fig. 3) was designed as a building block for a six-component one-pot assembly of **9**. However, the efforts failed due to the migration of levulinoyl group from the $O$-5 to $O$-6 position. As an alternative route, disaccharide **24b** was synthesized (Supplementary Fig. 3). Therefore, a four-component one-pot coupling reaction ($\mathbf{18+24b+24b+19}$) finally gave the

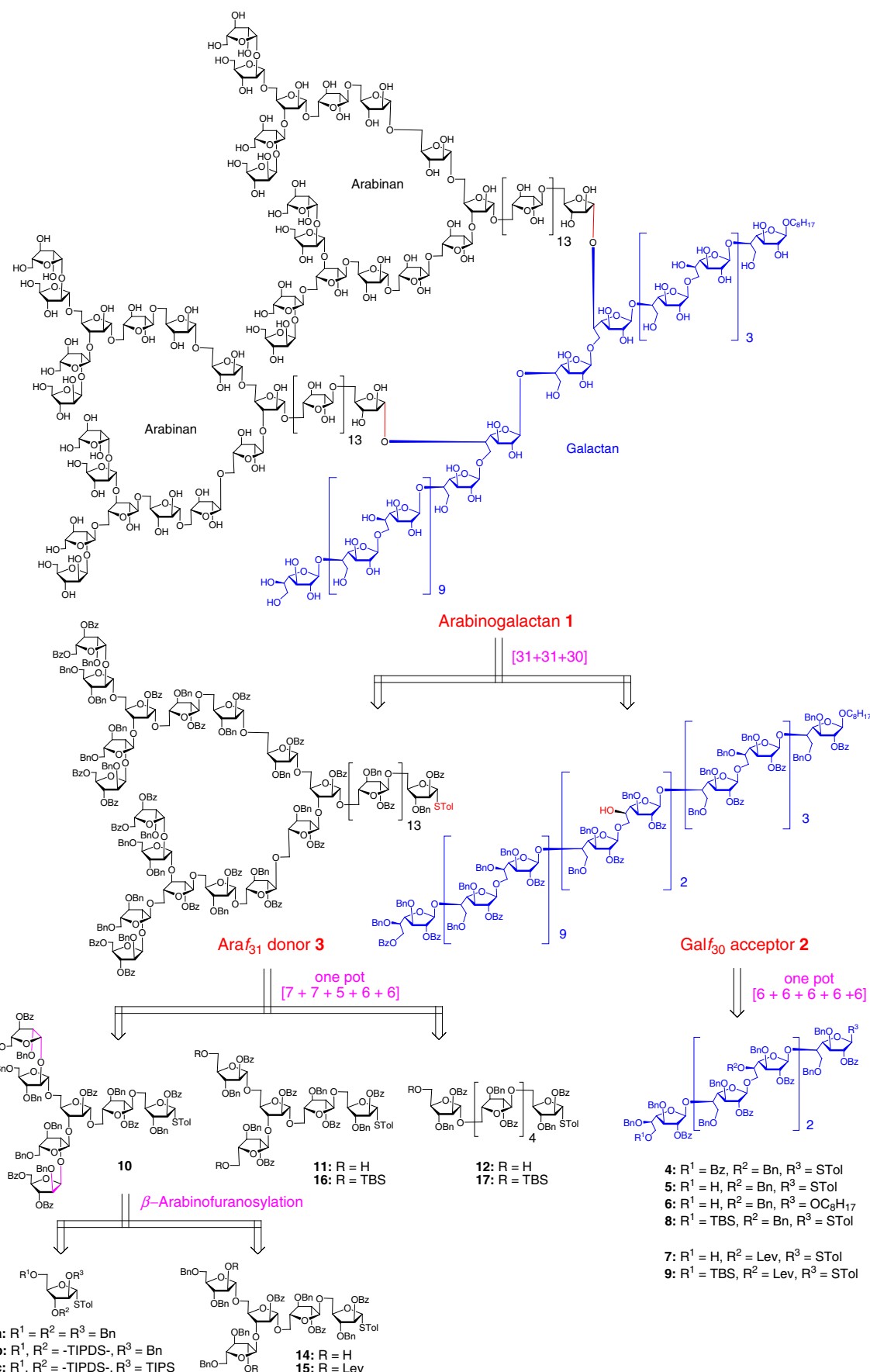

**Figure 1 | The structure of mycobacterial arabinogalactan 1 and its retrosynthetic analysis.** Bn, benzyl; Bz, benzoyl; Lev, levulinoyl; TBS, *tert*-butyl-dimethylsilyl; TIPDS, tetraisopropyldisiloxanylidene; TIPS, triisopropylsilyl; Tol, *p*-tolyl.

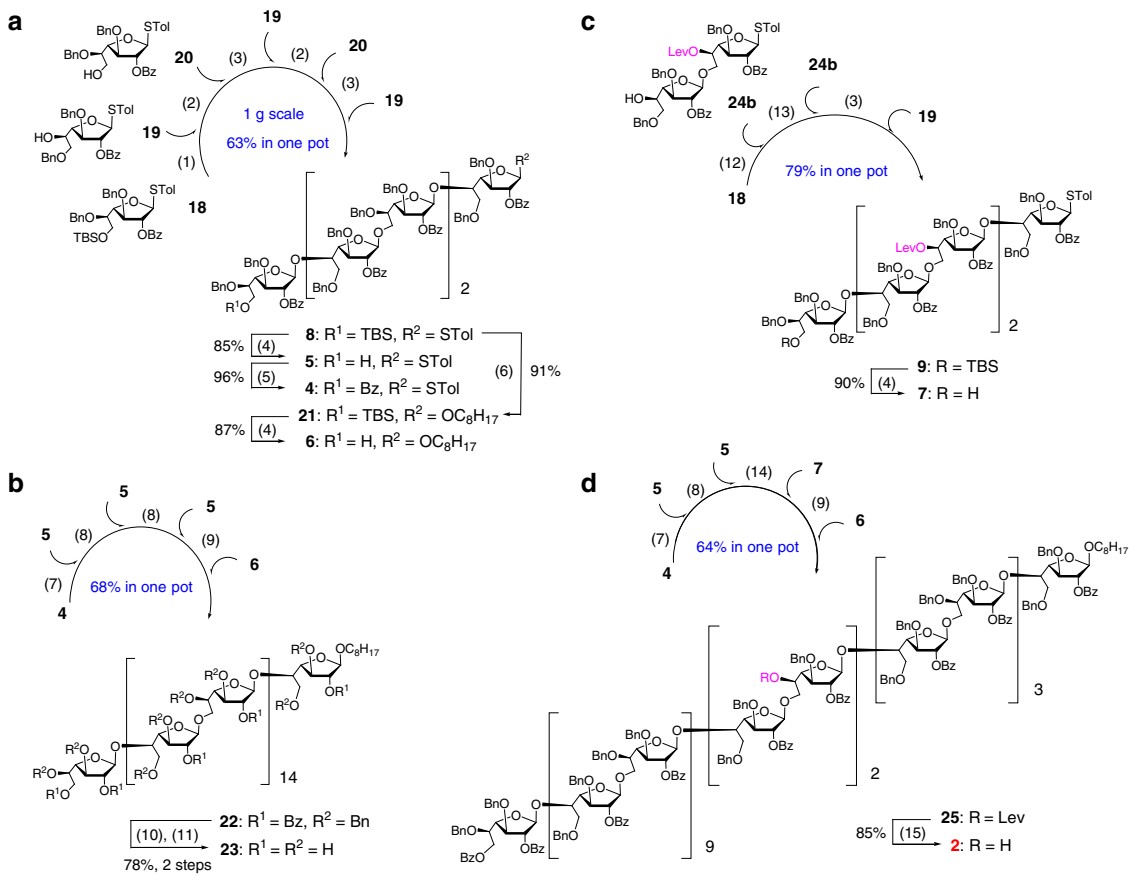

**Figure 2 | Synthesis of Gal*f*₃₀ acceptor 2.** (**a**) Synthesis of oligosaccharides **4**–**6**; (**b**) Synthesis of 30-mer galactan **23**; (**c**) Synthesis of hexasaccharide **7**; (**d**) Synthesis of Gal*f*₃₀ acceptor **2**. Reagents and conditions: (1) TTBP, 4 Å MS, CH₂Cl₂, *p*-TolSCl, AgOTf, then **19**, −78 °C to room temperature; (2) *p*-TolSCl, AgOTf, then **20**, −78 °C to room temperature; (3) *p*-TolSCl, AgOTf, then **19**, −78 °C to room temperature; (4) HF-pyridine, THF/H₂O (10:1), 35 °C; (5) Bz₂O, DMAP, pyridine, CH₂Cl₂, reflux; (6) *p*-TolSCl, AgOTf, TTBP, 1-octanol, 4 Å MS, CH₂Cl₂, −78 °C; (7) TTBP, 4 Å MS, CH₂Cl₂, *p*-TolSCl, AgOTf, then **5**, −78 °C to room temperature; (8) *p*-TolSCl, AgOTf, then **5**, −78 °C to room temperature; (9) *p*-TolSCl, AgOTf, then **6**, −78 °C to room temperature; (10) NaOCH₃, CH₃OH/CH₂Cl₂ (2:1); (11) Pd/C, H₂, EtOAc/THF/1-PrOH/H₂O (2:1:1:1); (12) TTBP, 4 Å MS, CH₂Cl₂, *p*-TolSCl, AgOTf, then **24b**, −78 °C to room temperature; (13) *p*-TolSCl, AgOTf, then **24b**, −78 °C to room temperature; (14) *p*-TolSCl, AgOTf, then **7**, −78 °C to room temperature; (15) H₂NNH₂-AcOH, THF/CH₃OH (10:1). DMAP, 4,4-dimethylaminopyridine; MS, molecular sieves; TTBP, 2,4,6-tri-*tert*-butylpyrimidine.

desired hexasaccharide **9** in 79% yield (Fig. 2c), which was subjected to desilylation to afford **7** (90% yield). Subsequently, polysaccharide **25** was assembled in a five-component one-pot manner as described in the construction of **22** by coupling the oligosaccharide fragments (**4** + **5** + **5** + **7** + **6**) in 64% overall yield (Fig. 2d). Exposure of **25** to hydrazine acetate successfully fulfilled the preparation of the desired Gal*f*₃₀ acceptor **2** (85% yield).

**Synthesis of Ara*f*₃₁ donor 3.** The assembly of the branched Ara*f*₃₁ donor **3** required three oligosaccharide intermediates, that is, β-Ara*f*-containing heptasaccharide **10**, branched pentasaccharide **11** and linear hexasaccharide **12**. For this purpose, a set of arabinofuranosyl building blocks (**13a–c**, **26–29**) were designed and synthesized (Supplementary Fig. 4). A six-component iterative one-pot glycosylation of monosaccharides **26** and **27** (**26** + **27** + **27** + **27** + **27** + **27**) afforded hexasaccharide **17** in excellent yield (73%) and on gram scale (1.20 g) (Fig. 3a). The desilylation of **17** resulted in the desired hexasaccharide **12** (92% yield). Likewise, the one-pot coupling reaction of building blocks **26**, **28** (ref. 30) and **27** provided a branched pentasaccharide **16** very smoothly (78% yield), which was further converted into diol **11** by desilylation in 95% yield (Fig. 3b). For the preparation of heptasaccharide **10**, another diol

**14** was required. Initially, the glycosyl donor **29a** (ref. 31) with chloroacetyl group at the *O*-2 position was chosen for the one-pot construction of pentasaccharide **15a** (Supplementary Table 1), but the overall yield was moderate (43%). Fortunately, when the donor **29** equipped with a levulinoyl group was employed for the one-pot glycosylation reaction, pentasaccharide **15** was rapidly assembled in 76% overall yield (Fig. 3b). Ultimately, deacylation of **15** gave diol **14** (94% yield).

Our attention was then turned to the synthesis of heptasaccharide **10**, which involved the stereocontrolled installation of two challenging β-arabinofuranosyl linkages. Among a number of innovative glycosyl donors developed by several groups[25,31–37], perbenzyl-protected thioglycoside **13a** (ref. 25) and 3,5-*O*-tetraisopropyldisiloxanylidene-protected thioglycosides **13b,c** (ref. 31) were synthesized for the current purpose (Supplementary Table 2). Although these donors proved useful for direct β-arabinofuranosylation, whether they could be subjected to thioglycoside acceptor **14** under the donor-preactivation conditions[38] remained to be explored. After some optimization, the best result arose when 4.0 equiv. of **13b** was preactivated by *p*-TolSCl/AgOTf and subsequently glycosylated with 1.0 equiv. of diol **14**, delivering heptasaccharide **30b** with good stereoselectivity (β,β-isomer/other isomers = 9/1). Removal of the silyl groups in **30b** afforded **31** (74% over two steps, Fig. 3c), in which the newly formed β-arabinofuranosyl linkages were

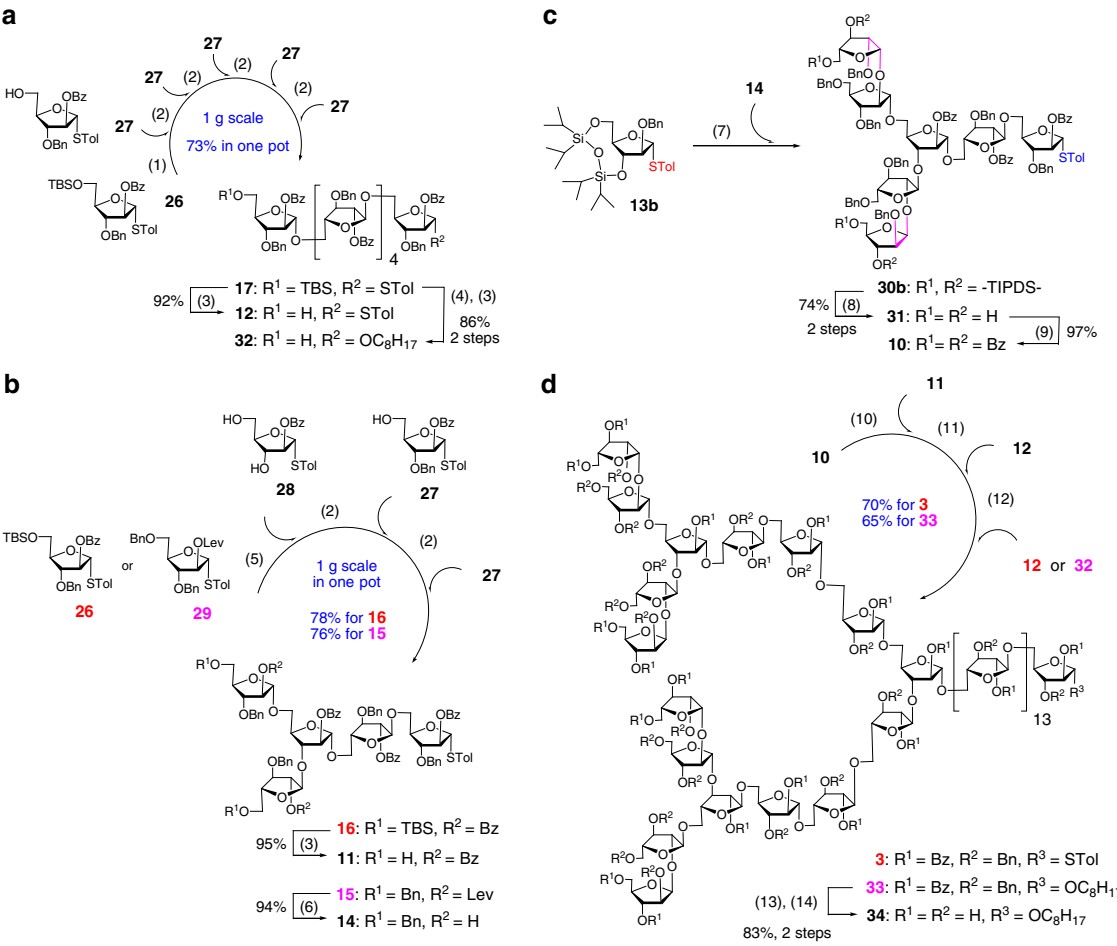

**Figure 3 | Synthesis of Araf₃₁ donor 3. (a)** Synthesis of oligosaccharides **12** and **32**; (**b**) Synthesis of diols **11** and **14**; (**c**) Synthesis of heptasaccharide **10**; (**d**) Synthesis of Araf₃₁ donor **3** and 31-mer arabinan **34**. Reagents and conditions: (1) TTBP, 4 Å MS, CH₂Cl₂, p-TolSCl, AgOTf, then **27**, −78 °C to room temperature; (2) p-TolSCl, AgOTf, then **27**, −78 °C to room temperature; (3) TBAF, AcOH, THF; (4) p-TolSCl, AgOTf, TTBP, 1-octanol, 4 Å MS, CH₂Cl₂, −78 °C; (5) TTBP, 4 Å MS, CH₂Cl₂, p-TolSCl, AgOTf, then **28**, −78 °C to room temperature; (6) H₂NNH₂·AcOH, THF/CH₃OH (10:1); (7) p-TolSCl, AgOTf, then **14**, −78 °C; (8) TBAF, THF; (9) Bz₂O, DMAP, pyridine, CH₂Cl₂, reflux; (10) TTBP, 4 Å MS, CH₂Cl₂, p-TolSCl, AgOTf, then **11**, −78 °C to room temperature; (11) p-TolSCl, AgOTf, then **12**, −78 °C to room temperature; (12) p-TolSCl, AgOTf, then **12** or **32**, −78 °C to room temperature; (13) NaOCH₃, CH₃OH/CH₂Cl₂ (2:1); (14) Pd/C, H₂, EtOAc/THF/1-PrOH/H₂O (2:1:1:1). TBAF, tetra-n-butylammonium fluoride.

confirmed by the $^{13}$C NMR spectrum (appearance at 99.6 and 99.1 p.p.m.)[39]. Finally, the re-protection of **31** with benzoyl groups yielded the desired heptasaccharide **10** (97% yield).

With oligosaccharide building blocks **10–12** in hand, the assembly of Araf₃₁ donor **3** by preactivation-based one-pot glycosylation protocol was attempted. This one-pot reaction was expected to be more challenging due to the steric hindrance in the double glycosylation of Araf₅ acceptor **11** using Araf₇ donor **10**. Surprisingly, the reaction proceeded smoothly when 2.3 equiv. of **10** was reacted with 1.0 equiv. of **11**, delivering an Araf₁₉ intermediate, which was sequentially coupled with two Araf₆ acceptors **12** in a single flask without any intermediate isolation to afford the Araf₃₁ donor **3** in 70% overall yield (Fig. 3d). To further confirm the identity of this 31-mer polysaccharide, an Araf₆ acceptor **32** bearing an alkyl group at the reducing end was synthesized (Fig. 3a). Thus a four-component one-pot coupling reaction of oligosaccharides **10–12** and **32** gave a similar 31-mer polysaccharide **33** in 65% yield (Fig. 3d), which was fully deprotected via deacylation and hydrogenolysis to afford the arabinan **34** ([M + Na]$^+$ monoisotopic m/z calcd. for 4246.4, found: 4246.3; [M + K]$^+$ monoisotopic m/z calcd. for 4262.4, found: 4262.2). Gratifyingly, the $^1$H and $^{13}$C NMR data of **34** were found to be identical

with previous reports[14,40] except for the differences in some repeating units.

**Assembly of arabinogalactan 1.** Our final task was the glycosylation of Galf₃₀ acceptor **2** with Araf₃₁ donor **3** to finish the assembly of target polysaccharide. To the best of our knowledge, no glycosylation reactions between polysaccharide sequences composed of >20 units were reported to date. For the planned [31 + 31 + 30] coupling reaction, it was anticipated that the biggest challenge would come from the steric hindrance by the bulky size of both the donor and acceptor, especially when a double glycosylation was required. Indeed, when a wide variety of promoter systems such as p-TolSCl/AgOTf[29], NIS/AgOTf[41], NIS/TfOH[42], N-(p-methylphenylthio)-ε-capro-lactam/Tf₂O[43], TBPA[44], Ph₃Bi(OTf)₂ (ref. 45), BSP/Tf₂O[46] and Ph₂SO/Tf₂O[47] were examined (Supplementary Table 3), no double glycosylation product or only some monoglycosylation product was observed before the donor decomposed, prompting us to further screen the reaction conditions. Encouragingly, it was found that benzenesulfinyl morpholine/triflic anhydride (BSM/Tf₂O)[48] developed by our group is the most effective promoter. And indeed, when promoted by BSM/Tf₂O, this double

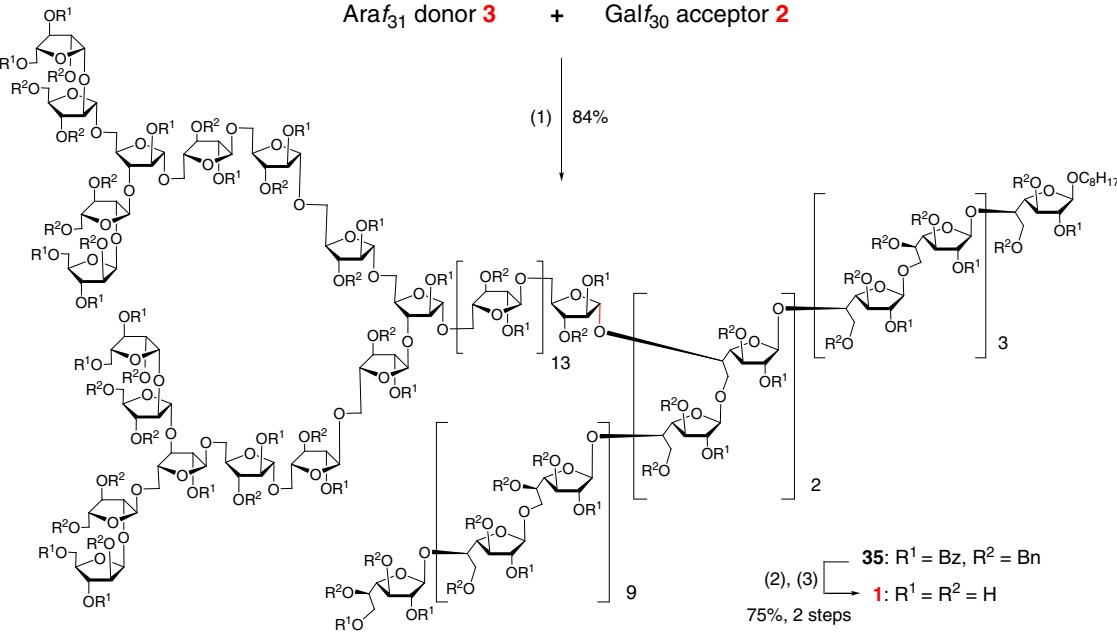

**Figure 4 | Assembly of arabinogalactan 1.** Reagents and conditions: (1) BSM, Tf$_2$O, 4 Å MS, CH$_2$Cl$_2$, − 40 °C; (2) NaOCH$_3$, CH$_3$OH/THF (2:1); (3) Pd/C, H$_2$, EtOAc/THF/1-PrOH/H$_2$O (2:1:1:1). BSM, benzenesulfinyl morpholine.

glycosylation was extremely clean and complete (indicated by thin-layer chromatography analysis), delivering the fully protected arabinogalactan **35** in 84% yield (Fig. 4). Although signals of the anomeric protons in $^1$H NMR spectrum were obscured due to the extensive overlapping, the anomeric carbons were distinctive in $^{13}$C NMR spectrum (all anomeric carbons of α-Araf residues and β-Galf residues were between 105 and 107 p.p.m., and anomeric carbons of β-Araf residues appeared at 101.0 and 100.6 p.p.m.). The identity of **35** was further supported by its MALDI-TOF mass spectrum ($[M + Na]^+$ $m/z$ calcd. for 33885.4, found: 33884.7). Finally, the global deprotection of **35** by the successive deacylation and hydrogenolysis was conducted, affording the target polysaccharide arabinogalactan **1** successfully.

## Discussion

We have developed a concise and highly efficient strategy for the first total synthesis of 92-mer mycobacterial arabinogalactan **1**. This work not only represents the longest well-defined carbohydrate chain synthesis up to date, but also provides useful compounds as probes for further investigations on mycobacterial cell wall-related biological events. Our synthetic strategy highlights a series of efficient preactivation-based one-pot glycosylation reactions to minimize the synthetic steps, the stereoselective β-arabinofuranosylation by preactivation protocol and the convergent [31 + 31 + 30] double glycosylation reaction, thus offering a straightforward access to the target polysaccharide. Our work may open an avenue to the synthesis of complex polysaccharides with biological importance that are either difficult or impossible to access through isolation or semisynthesis.

## Methods

**General.** The complete experimental details and compound characterization data can be found in Supplementary Methods. For the NMR, HPLC and MALDI-TOF mass spectra of the compounds in this article, see Supplementary Figs 5–126.

**General procedure for preactivation-based one-pot glycosylation reaction.** A mixture of glycosyl donor, TTBP and freshly activated 4 Å molecular sieves in anhydrous CH$_2$Cl$_2$ under argon atmosphere was stirred for 20 min at room temperature and cooled to − 78 °C. After 5 min, stoichiometric amount of p-TolSCl was added to the mixture, followed by the addition of AgOTf. After another 5 min, a solution of glycosyl acceptor in anhydrous CH$_2$Cl$_2$ was slowly added. The resulting mixture was slowly warmed to room temperature within 2 h, stirred for another 20 min and cooled back to − 78 °C. The glycosylation operation mentioned above was repeated until the generation of the desired product.

**Data availability.** The authors declare that the data supporting the findings of this study are available within the article and its Supplementary Information files. And all data are available from the authors upon reasonable request.

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

## Acknowledgements

This work was financially supported by the grants from the National Natural Science Foundation of China (21232002) and the Ministry of Science and Technology of China (2013CB910700, 2012CB822100). We thank Professor Qin Li and Professor Lijun Zhong at Peking University Health Science Center for their helpful assistance in analysis of glycan structures.

## Author contributions

D.-C.X. and X.-S.Y. conceived the research. Y.W., D.-C.X. and X.-S.Y. designed the experiments. Y.W. performed all the experiments. S.-C.C. and Y.-S.W. synthesized some building blocks. Y.W., D.-C.X. and X.-S.Y. analyzed the data. Y.W. and X.-S.Y. wrote the manuscript. X.-S.Y. supervised the project.

## Additional information

**Competing interests:** The authors declare no competing financial interests.

**Publisher's note**: 

