## [Peer review file · Nature Communications]

Reviewers' comments:

Reviewer #1 (Remarks to the Author):

The manuscript by Ye and co-workers describes the author's investigations into the chemical synthesis of complex mycobacterial arabinogalactan polysaccharide using a preactivation-based one-pot glycosylation approach. The methodology was applied to the synthesis of an arabinogalactan containing 92 monosaccharide units which the authors correctly identify as the longest homogeneous synthetic oligosaccharide reported to date. The work described in this manuscript clearly builds on earlier reports of related oligosaccharides but the current work is marked by the impressive size and complexity of the oligosaccharides that are reported. The scale on which the synthesis was conducted is also noteworthy. The reported work will be of considerable interest to the synthetic carbohydrate community and to those working in the bacterial oligosaccharide field. Overall the research is well described and the supporting information is sufficient to demonstrate that the target product was indeed prepared, however, there are some concerns regarding the yield of the deprotection step. It is the opinion of this referee that the manuscript is suitable for publication in Nature Communications subject to major revisions.

1. The manuscript requires significant proof reading and revision to improve the quality of English. For example, sentences in the abstract such as 'making carbohydrate synthesis to evolve slowly' and 'we challenge the carbohydrate synthesis' need to be completely restructured. These types of errors occur throughout the manuscript and will need to be addressed.
2. The reported yield of the deprotection step is of some concern since this is one of the most important and challenging steps in the synthesis. The authors report a yield of 13.1 mg or 75% over the two deprotecting steps. Since they started with 15mg of protected compound 35 and the mw of the deprotected compound is significantly lower (the exact mw is not quoted in the SI) it is not clear how the authors could have isolated 13.1 mg of product? This point would need to be clarified prior to publication.

Reviewer #2 (Remarks to the Author):

In this manuscript, the chemical synthesis of mycobacterial arabinogalactan with 92 sugar residues has been achieved. This is going to be the longest ever complex oligosaccharide synthesized by any human being thus far. The complexities and intricacies involved in this 92-mer synthesis are extremely difficult and Ye group achieved this feat with clinical precision. These oligosaccharide fragments will be of immense importance to the mycobacteriologists for the development of a vaccine against tuberculosis. Most of the chemistry to achieve this molecule is already well explored and developed. The novelty of this paper lies in the fact of daring to dream and executing that dream to reality. Chemical synthesis of oligosaccharides is still a challenging task. Realizing a 92-mer is a very bold step and Ye's group shall be acclaimed for the same. Here are some of the suggestions:

1. Figure 2 and in others as well: authors write that they performed the reactions at 1 g scale and in a one pot manner

NO supporting data was provided for the progress of the individual reactions of the one pot. It is also not clear how authors have achieved 63% yield in 5 or 6 steps. This clearly means that all the reactions occurred in more than 90% yield. It is understood that -STol is a fantastic donor; however, what about the side products and their interference in the reaction? On a whole the reaction towards the end of the 4th step will have more than 5 g of each reagent that is used. Are they not interfering in the reaction? Explanation towards this is highly warranted in the manuscript and the supporting information.

2. Reaction profiles by LC-MS or HPLC of crude reaction mixture at the end of each step of the one-pot reaction sequence shall be provided for all the reactions. In the absence of such kind of profiles, the claim of one-pot synthesis makes no sense.

3. The deprotection of compound 35, check the yield of reaction. One cannot get 13.1 mg of compound 1 from 15 mg of compound 35 as a large number of benzoyl esters and benzyl ethers are deprotected which significantly reduces the amount of material. Check and correct it in SI/Manuscript

4. Authors should mention: how did they assign overlapping carbon and proton resonances of compound 8 and many others for individual rings/sugars/carbons/hydrogens as they are all crowded. The method of assigning peaks shall be mentioned/illustrated in the SI.

5. The manuscript requires a lot of improvement in terms of language - Writing in the abstract: "We challenge carbohydrate synthesis" means what. This sentence has been repeated in the manuscript. There are many grammatical, linguistic and spelling mistakes in the manuscript. These are to be rectified.

Overall, this manuscript is going to be a landmark paper in the annals of oligosaccharide synthesis. I strongly recommend its publication in the Nature Communications after the above suggestions are addressed.

Reviewer #3 (Remarks to the Author):

The manuscript by Ye and co-workers describes the synthesis of a 91-mer mycobacterial arabinoglactan using a one pot [30+30+31] glycosylation reaction. All of the fragments used to assemble the backbone are also derived from one-pot syntheses, allowing for a highly convergent approach to target backbone. The synthesis takes great advantage of the fact that the vast majority of the linkages in the target (except for 8 terminal Araf residues) are 1,2-trans linked, which allows for neighboring group participation to control selectivity in the reaction. As with most one-pot approaches the more difficult 1,2-cis araf residues must be installed using a more standard solution phase approach. Nevertheless, this is a very impressive achievement considering the difficulty in coupling large oligosaccharide fragments. The novelty of the work lies in the sheer size of both the target molecule and the use of a one-pot approach to assemble very large (greater than 30 residue) fragments together. It also inadvertently highlights another issue with oligosaccharide synthesis, simply with structures this large it can be extremely difficult (if not impossible) to determine if the couplings were entirely stereoselective and it has to be taken on faith that neighboring group participation was able to exert absolute stereocontrol over the course of the reaction (for example, if the final anomeric ratio was 10:1 or 20:1 it would be next to impossible to observe a minor anomer). That being said, however, his work is important and should be of great interest to the synthetic chemistry and chemical biology communities. As such it should be acceptable for Nature Communications, once the following issues have been addressed.

For the one-pot procedures, tables detailing the optimization of the reaction are referenced in the text, but written in the SI. If space permits this data should be placed in the main text. For the final coupling the authors note that only BSM/Tf₂O was an effective promoter. This appears a little odd considering that other promoters (BSP/Tf₂O, PhSO₂/Tf₂O) failed completely. The authors should comment on this, did the reagents lead to decomposition, or was only monoglycosylation observed, or no reaction? This is especially true of entry nine where the reaction was stopped after 6 minutes (whereas most reactions were run for 10 hours). Such information would be very useful for other experimentalists planning similar routes.

Finally there are a few typos throughout the manuscript. For example stereo-selective should be stereoselective. The authors should carefully proof the manuscript.

Reviewer #1:

Comment 1: The manuscript requires significant proof reading and revision to improve the quality of English. For example, sentences in the abstract such as ‘making carbohydrate synthesis to evolve slowly’ and ‘we challenge the carbohydrate synthesis’ need to be completely restructured. These types of errors occur throughout the manuscript and will need to be addressed.

Answer: Thank you for your kind suggestions. As you suggested, we have deleted the sentences “making carbohydrate synthesis to evolve slowly” and “we challenge carbohydrate synthesis”. These types of errors have been carefully revised and the language has been improved.

Comment 2: The reported yield of the deprotection step is of some concern since this is one of the most important and challenging steps in the synthesis. The authors report a yield of 13.1 mg or 75% over the two deprotecting steps. Since they started with 15mg of protected compound **35** and the mw of the deprotected compound is significantly lower (the exact mw is not quoted in the SI) it is not clear how the authors could have isolated 13.1 mg of product? This point would need to be clarified prior to publication.

Answer: Thanks for your careful reading of our manuscript. The expression in our synthetic procedure is probably unclear. Indeed, we started the deprotection step with 15 mg of compound **35**, but this deprotection process was repeated twice. That is, totally 3 individual deprotection reactions were performed, thus a total amount of 3 X 15 mg (45 mg) of compound **35** was used, providing 13.1 mg of final product in 75% isolated yield. This is evidenced by the previous statement “This global deprotection process was repeated twice, and the combined crude products were purified by gel filtration”. We are sorry about our unclear expression. To make a clear expression, we have added a statement “The global deprotection was started with 45.0 mg of fully protected **35**, which was divided into 3 portions (15.0 mg each portion) to carry out 3 individual reactions” to the revised SI (see Page S199).

Reviewer #2:

Comment 1: Figure 2 and in others as well: authors write that they performed the reactions at 1 g scale and in a one pot manner

NO supporting data was provided for the progress of the individual reactions of the one pot. It is also not clear how authors have achieved 63% yield in 5 or 6 steps. This clearly means that all the reactions occurred in more than 90% yield. It is understood that -STol is a fantastic donor; however, what about the side products and their interference in the reaction? On a whole the reaction towards the end of the 4th step will have more than 5 g of each reagent that is used. Are they not interfering in the

reaction? Explanation towards this is highly warranted in the manuscript and the supporting information.

Answer: Thank you very much for your reviewing our manuscript. As illustrated in the following scheme, when the donor **18** was pre-activated by stoichiometric amount of *p*-TolOTf and subsequently reacted with the thioglycoside acceptor **19** to generate a disaccharide intermediate, there are typically three kinds of side products: disulfide, glycal, and TfOH (it can be scavenged by TTBP), and none of these side products interfered the glycosylation reaction based on our observation during the one-pot operation. Thus, our one-pot glycosylation reaction is highly efficient. The TLC monitoring the whole one-pot process of Fig. 2a (see the response to comment 2 below) provides the further proof. It is noteworthy that all the building blocks and reagents should be added stoichiometrically to ensure a successful one-pot glycosylation in satisfactory yield. Due to the space limit, we added one sentence to the revised manuscript (see Page 4). We have detailed one-pot experimental procedures for the preparation of oligosaccharides in the SI.

Comment 2: Reaction profiles by LC-MS or HPLC of crude reaction mixture at the end of each step of the one-pot reaction sequence shall be provided for all the reactions. In the absence of such kind of profiles, the claim of one-pot synthesis makes no sense.

Answer: The reaction profile by LC-MS or HPLC of crude reaction mixture is just one of choice for the profile of one-pot synthesis. In our experiments, we monitored each step of the one-pot reaction sequence by TLC detection, because TLC monitoring is more convenient and the cost is cheaper. Herein, as exemplified, two sets of TLC profiles as the proof to support the whole process of one-pot reaction were provided as follows:

1) One-pot synthesis of hexasaccharide **8** (the original record is also attached)

TLC Profile of one-pot synthesis of hexasaccharide **8**

Original record of one-pot synthesis of hexasaccharide **8**

2) One-pot synthesis of 30-mer polysaccharide **22** (the original record is also attached)

TLC Profile of one-pot synthesis of 30-mer polysaccharide **22**

Original record of one-pot synthesis of polysaccharide **22**

As shown in the TLC profiles, during one-pot glycosylation reactions, for each coupling step, one major spot can be seen. That means the desired oligosaccharide can be isolated in good yield.

Comment 3: The deprotection of compound 35, check the yield of reaction. One cannot get 13.1 mg of compound 1 from 15 mg of compound 35 as a large number of benzoyl esters and benzyl ethers are deprotected which significantly reduces the amount of material. Check and correct it in SI/Manuscript

Answer: Thanks for your careful reading of our manuscript. The expression in our synthetic procedure is probably unclear. Indeed, we started the deprotection step with 15 mg of compound **35**, but this deprotection process was repeated twice. That is, totally 3 individual deprotection reactions were performed, thus a total amount of 3 X 15 mg (45 mg) of compound **35** was used, providing 13.1 mg of final product in 75% isolated yield. This is evidenced by the statement “This global deprotection process was repeated twice, and the combined crude products were purified by gel filtration”. We are sorry about our unclear expression. To make a clear expression, we have added a statement “The global deprotection was started with 45.0 mg of fully protected **35**, which was divided into 3 portions (15.0 mg each portion) to carry out 3 individual reactions” to the revised SI.

Comment 4: Authors should mention: how did they assign overlapping carbon and proton resonances of compound 8 and many others for individual rings/sugars/carbons/hydrogens as they are all crowded. The method of assigning peaks shall be mentioned/illustrated in the SI.

Answer: Thanks for your comments. Indeed, we tried our best to assign the characteristic proton and carbon signals of oligosaccharides synthesized in this manuscript (especially those on the anomeric center) using 600 MHz ¹H-¹H COSY, HSQC and HMBC experiments, which has been mentioned in the General methods of SI (see Page S129).

Comment 5: The manuscript requires a lot of improvement in terms of language - Writing in the abstract: "We challenge carbohydrate synthesis" means what. This sentence has been repeated in the manuscript. There are many grammatical, linguistic and spelling mistakes in the manuscript. These are to be rectified.

Answer: Thanks for your suggestions. We have deleted the statement “We challenge carbohydrate synthesis”, and the grammatical, linguistic and spelling mistakes in the manuscript have been carefully checked and corrected.

Reviewer #3:

Comment 1: For the one-pot procedures, tables detailing the optimization of the reaction are referenced in the text, but written in the SI. If space permits this data should be placed in the main text.

Answer: Thank you very much for your reviewing our manuscript. Because we've got 4 large Figures in the main text, especially Figure 1 which outlines the retrosynthetic analysis of the target compound, so we are afraid that no additional space is available, we therefore place tables detailing the optimization of the one-pot reactions in the SI.

Comment 2: For the final coupling the authors note that only BSM/Tf₂O was an effective promoter. This appears a little odd considering that other promoters (BSP/Tf₂O, PhSO₂/Tf₂O) failed completely. The authors should comment on this, did the reagents lead to decomposition, or was only monoglycosylation observed, or no reaction? This is especially true of entry nine where the reaction was stopped after 6 minutes (whereas most reactions were run for 10 hours). Such information would be very useful for other experimentalists planning similar routes.

Answer: The suggestions are gratefully appreciated. For the final coupling reaction, we found that only promoter BSM/Tf₂O gave the best results. The use of BSP/Tf₂O led to decomposition of the donor and no coupled product was detected by TLC monitoring. When promoted by Ph₂SO/Tf₂O, only some monoglycosylation product was observed. As for entry 9, since the reaction was performed at room temperature, we stopped the reaction after 6 min because no desired product was formed before the donor decomposed. The reaction time is generally few minutes when promoted by *N*-(phenylthio)- ϵ -caprolactam/Tf₂O at room temperature, which is evidenced by Wong and coworkers (Duroń, S. G., Polat, T. & Wong, C.-H., *Org. Lett.* **2004**, *6*, 839-841). We have revised the statement in the revised main text (see Page 7). Also, we added a footnote to Supplementary Table 3 in the SI.

Comment 3: Finally there are a few typos throughout the manuscript. For example stereo-selective should be stereoselective. The authors should carefully proof the manuscript.

Answer: Thank you for your careful reading of our manuscript. We have changed 'stereo-selective' into 'stereoselective'. In addition, this manuscript has been carefully revised.

REVIEWERS' COMMENTS:

Reviewer #1 (Remarks to the Author):

The manuscript by Ye and co-workers describes the author's investigations into the chemical synthesis of complex mycobacterial arabinogalactan polysaccharide using a preactivation-based one-pot glycosylation approach. The authors have satisfactorily addressed the comments provided in the review and it is the opinion of this referee that the manuscript is suitable for publication in Nature Communications.

Reviewer #2 (Remarks to the Author):

All suggestions were incorporated. The manuscript is now suitable for publication in Nature Communications

Reviewer #1, in comments to the editor, asked for HPLC or LC-MS spectra for compound 1 to be provided.

Answer: As you suggest, we provide a HPLC spectrum for compound 1 as follows (please see Supplementary Figure 126 in the Supplementary Information):

Supplementary Figure 126. HPLC trace of compound 1. Instruments: Shimadzu LC-10AT liquid chromatograph equipped with ELSD detector (ELSD 2000ES); Conditions: Waters XBridge[®] C18 5 μ m, 4.6 \times 250 mm column, 0-30 min linear gradient: 5-95% CH₃CN, H₂O, 1 mL/min flow.